# Periodontitis and Gestational Diabetes Mellitus: A Potential Inflammatory Vicious Cycle

**DOI:** 10.3390/ijms222111831

**Published:** 2021-10-31

**Authors:** María José Bendek, Gisela Canedo-Marroquín, Ornella Realini, Ignacio N. Retamal, Marcela Hernández, Anilei Hoare, Dolores Busso, Lara J. Monteiro, Sebastián E. Illanes, Alejandra Chaparro

**Affiliations:** 1Department of Periodontology, Centre for Biomedical Research, Faculty of Dentistry, Universidad de los Andes, Av. Plaza 2501, Las Condes, Santiago 7620157, Chile; mbendek@miuandes.cl (M.J.B.); gise.canedo@gmail.com (G.C.-M.); or.realini@gmail.com (O.R.); iretamal@miuandes.cl (I.N.R.); 2Laboratory of Periodontal Biology and Department of Pathology and Oral Medicine, Faculty of Dentistry, University of Chile, Olivos 943, Independencia, Santiago 8380544, Chile; mhernandezrios@gmail.com; 3Laboratory of Oral Microbiology, Department of Pathology and Oral Medicine, Faculty of Dentistry, Universidad de Chile, Olivos 943, Independencia, Santiago 8380544, Chile; a.hoare@odontologia.uchile.cl; 4Program in Biology of Reproduction, Centre for Biomedical Research and Innovation (CIIB), Universidad de los Andes, Santiago 8380544, Chile; dbusso@uandes.cl (D.B.); lmonteiro@uandes.cl (L.J.M.); sillanes@uandes.cl (S.E.I.); 5IMPACT, Center of Interventional Medicine for Precision and Advanced Cellular Therapy, Santiago 8380544, Chile

**Keywords:** periodontitis, gestational diabetes mellitus, systemic inflammation, extracellular vesicles

## Abstract

Periodontitis is a chronic inflammatory immune disease associated with a dysbiotic state, influenced by keystone bacterial species responsible for disrupting the periodontal tissue homeostasis. Furthermore, the severity of periodontitis is determined by the interaction between the immune cell response in front of periodontitis-associated species, which leads to the destruction of supporting periodontal tissues and tooth loss in a susceptible host. The persistent bacterial challenge induces modifications in the permeability and ulceration of the sulcular epithelium, which facilitates the systemic translocation of periodontitis-associated bacteria into distant tissues and organs. This stimulates the secretion of pro-inflammatory molecules and a chronic activation of immune cells, contributing to a systemic pro-inflammatory status that has been linked with a higher risk of several systemic diseases, such as type 2 diabetes mellitus (T2DM) and gestational diabetes mellitus (GDM). Although periodontitis and GDM share the common feature of systemic inflammation, the molecular mechanistic link of this association has not been completely clarified. This review aims to examine the potential biological mechanisms involved in the association between periodontitis and GDM, highlighting the contribution of both diseases to systemic inflammation and the role of new molecular participants, such as extracellular vesicles and non-coding RNAs, which could act as novel molecular intercellular linkers between periodontal and placental tissues.

## 1. Introduction

Periodontitis is a multifactorial chronic inflammatory and destructive disease of the supporting periodontal tissues. It is characterized by a dysbiotic state and the disruption of the periodontal tissue homeostasis in a susceptible host, which clinically involves inflammation, clinical attachment loss, irreversible alveolar bone destruction, and increased risk of tooth loss [1]. The worldwide prevalence of periodontitis is challenging to estimate, since case definitions and differences in the study populations generate substantial heterogeneity among epidemiological studies [2]. The Global Burden of Disease Study 2013 calculated a worldwide number of cases of periodontal diseases worldwide of 503,967,200, and the Global Burden of Disease Study 2010 estimated a 10.8% prevalence of severe periodontitis, affecting 743 million people worldwide [3,4]. Thus, periodontitis represents a global public health concern because it has high prevalence, comorbidities, and economic burden. Moreover, periodontitis also affects health-related quality of life in terms of oral discomfort, pain, social and emotional alterations, and malnutrition [5,6,7].

The etiopathogenesis of periodontitis is described by the dysbiotic hypothesis, where pathobionts and specific keystone pathogens increase in proportion to symbiotic bacteria. These bacteria have immune subversion mechanisms, which provoke an inflammatory but inefficient immune response that encourages an increase in inflammophilic bacteria, triggering additional inflammatory responses and periodontal destruction in a dysbiotic state [8]. This chronic immune stimulation activates both the innate and adaptive immune systems. Moreover, the lymphocytic response elicits an increase in the local concentrations of receptor activator nuclear factor kappa B ligand (RANK-L). Consequently, osteoclast differentiation and bone reabsorption increase. These events lead to the formation of periodontal pockets, which favor the number of niches for the dysbiotic microbiome and which, in turn, act as a reservoir for inflammophilic bacteria, immune cells, and pro-inflammatory molecules [9,10,11]. Consequently, the ulceration and increased permeability of the junctional epithelium at periodontal pockets enhances the metastatic dissemination of periodontitis-associated bacteria, their related products (e.g., LPS), and pro-inflammatory molecules into the systemic circulation and long-distance tissues such as the fetal-maternal unit [12,13]. This phenomenon lays the foundations of the concept of “Periodontal Medicine” and usually occurs in daily activities (e.g., chewing or brushing one’s teeth) or after dental procedures (e.g., prophylaxis or subgingival debridement in periodontitis subjects) [14].

In the last decades, several studies have described the epidemiological associations between periodontitis and other non-communicable chronic inflammatory diseases, such as cardiovascular, respiratory, and metabolic afflictions, rheumatoid arthritis, Alzheimer’s disease, T2DM, and adverse pregnancy outcomes (APOs) [2,5,12,15,16,17,18,19,20,21]. Regarding the effects of periodontitis on pregnancy outcomes, most studies show an epidemiologic association between periodontitis and preterm birth, fetal growth restriction, preeclampsia (PE), and gestational diabetes mellitus (GDM) [22,23]. The biological plausibility for this association has been supported by (1) a direct mechanism that involves the translocation or metastatic dissemination of periodontal pathogens or their related byproducts to invade the fetal-placental unit via hematogenous spreading or in an ascending route from the genitourinary tract and (2) an indirect mechanism that is mediated by the pro-inflammatory mediators produced at inflamed periodontal tissues, which directly affects the placental tissue or impacts the synthesis of acute-phase molecules in the liver, such as CRP and IL-6 [12,23] (Figure 1).

In this review, we will discuss the potential biological mechanisms involved in the association between periodontitis and GDM, highlighting the contribution of both diseases to systemic inflammation and the role of new molecular participants, such as extracellular vesicles and non-coding RNAs, which could act as novel molecular intercellular linkers between periodontal and placental tissues

## 2. Gestational Diabetes Mellitus

Gestational diabetes mellitus (GDM) corresponds to glucose intolerance, with onset or first recognition during pregnancy [24]. The global prevalence of GDM ranges between 2% and 35% [25], and there is particular concern since this prevalence is increasing globally [26]. Moreover, with the new classification of GDM by the World Health Organization (WHO) and the American Diabetes Association (ADA), the global prevalence of GDM is expected to further increase by approximately twofold [27,28]. The main risk factors for GDM development are an aging population, urbanization, higher prevalence of obesity, sedentary lifestyles, and stressful modern lifestyles [30]. That aside, GDM pregnancies have a higher risk of developing hypertension and PE and a 50% chance of developing T2DM during the next 10–30 years [27].

Additionally, fetuses from GDM pregnancies are more susceptible to short-term adverse outcomes, such as macrosomia, neonatal hypoglycemia, neonatal cardiac dysfunction, and labor difficulties [29,30]. The hyperglycemic status in the intrauterine environment also influences the future newborn’s postnatal health through a phenomenon known as "fetal programming", which involves epigenetic changes, such as histone modifications and DNA methylation [31,32]. Fetal programming allows the fetus to adapt to pregnancy exposures; however, it can also predispose to the development of non-communicable chronic diseases, including cardiovascular disease [33,34], obesity, and T2DM [35]. Furthermore, females born from a GDM pregnancy have a higher chance of developing GDM during their pregnancies, creating a vicious cycle for this condition [24].

There is clinical evidence that the plasmatic levels of systemic markers of inflammation such as TNF-α, IL-6, and CRP are increased in GDM compared with euglycemic pregnant women [36,37]. In GDM pregnancies, circulating CRP levels are increased between the first and third trimester and maintained until 6 months after delivery compared with euglycemic controls [37,38,39,40]. It is believed that the source of pro-inflammatory cytokines in GDM may be the placenta and adipose tissue, but other tissues may also be involved [36]. GDM placentas have shown higher macrophage infiltration and mRNA expression of TNF-α, IL-6, and TLR-4 [36]. Moreover, obesity is another risk factor for GDM and, in that context, lipid accumulation increases the activation of NADPH oxidase and ROS production in adipocytes, inducing endoplasmic reticulum stress (ER stress), which activates the JNK pathway and induces insulin resistance [41]. However, the contribution of periodontitis to the systemic and placental inflammatory state in GDM conditions has not been uncovered yet. It is plausible that some of these molecules come from the inflamed periodontal tissues in periodontitis pregnancies, increasing the risk of developing GDM by an indirect mechanism (Figure 1). Although it is complex to identify the origin of plasmatic inflammatory mediators, a recent systematic review shows reductions of CRP and IL-6 in GCF and plasma after periodontal treatment in pregnant women [42].

Sustained hyperglycemia increases the damage-associated molecular patterns (DAMPs), like free fatty acids and glycation end products (AGEs), that activate the toll-like receptors (TLRs) on the cell surface, such as TLR-2 and TLR-4 in macrophages and placental cells [43]. The activation of TLRs generates the stimulation of the inflammasome within the cellular cytoplasm (NLRP-3), which activates caspase-1, which then cleaves the IL-1β pro-inflammatory cytokine, allowing its secretion. This pathway generates a pro-inflammatory response as a consequence of hyperglycemia that is codependent on the nuclear factor kappa-light-chain-enhancer of the activated B cells’ (NF-κB) signaling cascade [44]. Interestingly, periodontal bacteria or their bDNA are recognized by the same receptors in periodontal tissues, which are also stimulated by the hyperglycemic status (TLR-2, TLR-4, and TLR-9) [45,46]. Indeed, the repeated administration of *P. gingivalis* in a murine model led to an increased expression of pro-inflammatory cytokines and reduced insulin resistance in adipose tissue, as well as increased levels of TNF-α [47,48,49]. Furthermore, TLR-9 can modulate the response of TLR-2 and TLR-4 ligands, probably by cross-talking [50].

In addition, the placenta produces TNF-α and leptin, which are highly relevant in GDM pathogenesis since they induce insulin resistance in peripheric tissues and have been proposed as molecules involved in insulin resistance during pregnancy [51]. Maternal plasmatic concentrations of TNF-α and leptin are significantly higher in GDM, suggesting that the placenta is a source of these cytokines [52]. TNF-α induces insulin receptor substrate 1 (IRS-1) serine phosphorylation, downregulating insulin action and linking inflammation to insulin resistance in pregnancies [51]. Furthermore, leptin is considered an adipokine with pleiotropic effects. Leptin stimulates IL-6 and TNF-α secretion, oxidative burst, and the chemotactic response of immune cells [41]. The regulatory processes executed by the placenta are indirectly triggered by pro-inflammatory mediators, which are modulated by hyperglycemia, hyperlipidemia, and vascular oxidative stress, which are impaired in GMD [53].

Other placental hormones contribute to physiologic insulin resistance during normal pregnancies, but when the pancreatic compensatory insulin production is altered, as it is in GDM, they may end up contributing to GDM development. For instance, hPL in normal pregnancies enhances pancreatic insulin production when it binds its receptor in pancreatic beta cells through pancreatic and duodenal homeobox 1(PDX1) expression [54], although it has also been reported to act as an insulin antagonist [55,56], and mouse models lack the early expression of receptors for hPL generate obesity, hyperleptinemia and glucose intolerance [57]. Likewise, PLGF overexpression generates fasting and postprandial hyperinsulinemia with low glycemia reduction after insulin injection in transgenic mice [56]. That aside, PLGF protein expression is significantly higher in GDM placentas when compared with healthy controls [55].

## 3. Periodontitis and Gestational Diabetes Mellitus: Epidemiological Studies

Gingivitis is a highly prevalent condition during pregnancy (30–100%), with an increase in bleeding on probing (BOP) and gingival inflammatory signs such as redness and pseudo periodontal pockets. In addition, pre-pregnancy periodontitis could progress with further clinical attachment loss (CAL) (10–74% prevalence) [58]. Another clinical characteristic feature is the presence of pregnancy granulomas, known as Bloom’s pregnancy tumors (0.2–9.6% prevalence), which are not necessarily related with the amount of bacterial plaque [58]. It is believed that pregnancy granulomas are caused by the pro-angiogenic effect from higher progesterone and estradiol concentrations, although this cause has not been elucidated completely [59,60]. There are different hypotheses on the effect of steroid sex hormones in periodontal tissues: (1) they could cause an environmental change with an increase in the proliferation of *Prevotella spp*, which is highly controversial; (2) there is a pro-angiogenic effect of progesterone in the periodontium, which is related with an increase in the gingival crevicular fluid (GCF) efflux; (3) there is an increase in the secretion of interleukin-6 (IL-6) by immune cells stimulated by sex hormones; and (4) there is a proliferative effect from estradiol and progesterone in gingival fibroblast populations [61].

Current evidence suggests that pregnant women with severe periodontitis are at greater risk of developing GDM [62]. Indeed, multiple studies show an association between periodontitis and GDM, which is maintained after adjusting the confounding variables [63,64]. Moreover, the risk for PE development increased with periodontitis and GDM diagnosis by 20-fold (*p* = 0.001) [64]. In addition, systematic reviews agree that pregnant women with periodontitis have an increased risk for GDM after adjusting the confounding variables [65,66,67].

Regarding the association between periodontal clinical parameters and GDM, we observed significant increases in BOP (*p* = 0.003), PPD (*p* = 0.0028), CAL (*p* < 0.001), and periodontal inflamed surface area (PISA) (*p* < 0.001) in a prospective cohort study of pregnant women recruited at 11–14 gestation weeks who later developed GDM in comparison with women who experienced euglycemic pregnancies [68]. Furthermore, a study based on the National Health and Nutrition Examination Survey (NHANESIII) data showed a prevalence of 44.8% of periodontitis in GDM and 13.2% in euglycemic pregnant women, indicating a 9-fold increased risk for GDM pregnant women to develop periodontitis during their pregnancies (OR = 9.11, *p* < 0.05) [69].

A retrospective multivariate regression analysis of NHANES III reported that women who had a history of GDM, with or without current diabetes mellitus at the time of examination, were more likely to have periodontitis than women without a history of GDM without statistical significance, probably due to the subgroup sample size (Table 1) [70]. However, there are no prospective longitudinal studies that assess the bidirectionality of the association between periodontitis and GDM, and we propose that sustained hyperglycemia, as a standard feature of diabetes, could have an impact on the terms of periodontitis severity during pregnancy [21].

It is also relevant to consider obesity as a possible contributing factor between periodontitis and GDM association. A recent cross-sectional study reported higher PPD, CAL, and periodontitis prevalence in obese overweight compared with normal weight pregnant women (*p* = 0.041, 0.039, and <0.001, respectively) [71]. These results were ratified by a longitudinal study that compared periodontal health statuses between obese and non-obese pregnant women during their third trimesters and 2 months after delivery, where it was observed that obese pregnant women had higher PPD and CAL at both evaluation times and that a higher maternal BMI had a positive association with periodontitis prevalence during pregnancy (OR = 1.229; 95% CI: 1.1–1.38, *p* = 0.001) [72]. Therefore, epidemiological association studies between periodontitis and GDM ought to address obesity as a confounding variable.

Concerning the effect of periodontal treatment on GDM onset, a systematic review showed a significantly decreased risk of perinatal mortality but failed to find a significant association with GDM prevalence (RR = 1.60, 95% CI = 0.64–4.00; *p* = 0.31) [73]. However, this result is complex to interpret, because it included randomized clinical trials that were heterogeneous in terms of GDM diagnosis, periodontal treatment modality, and timing of intervention. Periodontal therapy is generally performed during the second trimester of pregnancy, almost at the same time as when GDM is diagnosed. Hence, the best timing to treat periodontitis would probably be before pregnancy or during early pregnancy to avoid the activation of an inflammatory state in a timely manner.

## 4. Periodontal Bacteria Translocation into the Maternal Fetal Unit as a Direct Mechanism of Association

Microbiological data suggest that periodontitis-associated bacteria, such as *Campylobacter rectus* (*C. rectus*), *F. nucleatum*, *P. gingivalis*, and *Bergeyella spp*. can cross the placenta [74]. In fact, periodontal bacteria may cause amniotic fluid infection, which is associated with preterm birth [75,76]. The intrauterine presence of *C. rectus* is related with retardation in fetal growth, impaired neurological development, and hypermethylation of fetal DNA in a mice model [13].

Among the periodontal bacteria involved in periodontitis and APO association, *Fusobacterium nucleatum* (*F. nucleatum*) and *Porphyromonas gingivalis* (*P. gingivalis*) are the most strongly associated with intrauterine infection [23]. The presence of *F. nucleatum* in the uterus can be associated with 10–30% of preterm births. It has been observed that *F. nucleatum* intravenous injection in pregnant mice leads to the colonization of placenta, preterm birth, and abortion [23]. Moreover, *F. nucleatum* aids the translocation of *P. gingivalis* into maternal tissues [77]. Studies suggest that *P. gingivalis* affects the immune host response during pregnancy with neutrophils and macrophage dysfunction, the release of pro-inflammatory cytokines, Th17’s shift of the T helper response, and an increase in the oxidative stress of the placental tissues [78,79]. Furthermore, LPS from *P. gingivalis* alters the function, proliferation, and apoptosis of extravillous trophoblasts, generating alterations in the maturation of the placenta and remodeling of the uterine spiral arteries, promoting endothelial dysfunction and inflammation [80,81].

Periodontal bacteria generate either direct or indirect effects within the mother and feto-maternal unit. There is also evidence that GDM patients have different oral microflora compositions, which may influence the glucose metabolism pathways. Thus, it is highly relevant to further study this topic to enrich the knowledge of GDM and periodontitis pathogenesis. For instance, Ganiger K. et al. observed that women with GDM had greater detection of *P. gingivalis* than healthy pregnant women by conventional PCR analysis from subgingival samples (*p* = 0.007) [82]. Li X. et al. recently reported ROC analysis for oral biomarkers of GDM with an AUC of 0.83 for a GDM prediction model characterized by the combination of *Lautropia* and *Neisseria* in dental plaque and *Streptococcus* in saliva [83]. In addition, other studies suggest a relationship between periodontal bacteria and vaginal microbiota such as *Prevotella intermedia* and *Porphyromonas endodontalis*, which were found to be increased in women with dysbiotic vaginal communities [84]. Vaginal microorganisms influence newborns’ birth canal bacterial colonization through delivery. It has been reported that neonates from GDM vaginally delivered pregnancies had different oral microflora compositions from the healthy controls, specifically with an increase in the major genera *Prevotella, Bacteroides, Bifidocaterium, Corynebacterium*, *Ureaplasma,* and *Weisella* (*p* < 0.05) [85].

## 5. Release of Pro-Inflammatory Molecules from Periodontal Inflamed Tissues and Systemic Inflammation as an Indirect Mechanism of Association

Periodontitis represents a constant systemic challenge for immunocompetent cells, with the activation of pro-inflammatory cascades that contribute to the systemic inflammatory status of the subject [86]. This is represented by an increase in acute phase response molecules, such as C-reactive protein (CRP) and pro-inflammatory cytokines (TNF-α and IL-6) [87,88]. However, the biological mechanisms and the molecules involved in the relationship between periodontitis and GDM are mostly unknown.

Our research group has also studied the presence of placental-like growth factor (PlGF) and matrix metalloproteinase-8 and -9 (MMP-8 and MMP-9) in the GCF from pregnant women, aiming to identify early risk predictors of GDM in oral fluids. Interestingly, GCF-PlGF levels were significantly higher in women who later were diagnosed with GDM (*p* = 0.0019). A multivariate model which included glycemia and GCF-PlGF during early pregnancy discriminated the future development of GDM with an AUC-ROC curve of 0.897 (78.6% sensitivity and 75% specificity) [68]. Moreover, MMP-8 and MMP-9 were found to have increased in the early stages of pregnancy (<14 weeks of gestation) in women who later developed GDM [89]. In the same study, clinical periodontal parameters were positively correlated with the expression of both MMPs in GCF, and pregnant women with periodontitis stage III and IV showed increased concentrations of MMP-8 when compared with pregnancies with periodontitis stage I [89]. Thus, increases in MMP-8 and MMP-9 levels in GCF were associated with both the severity of periodontitis and GDM development. MMPs could contribute to diabetes development by the degradation of insulin receptors [90], but the exact biological explanation for this association and the source of origin of MMPs remains to be clarified [91].

The hypothesis that the placental pro-inflammatory mediators’ production contributes to the “physiologic” low grade systemic inflammation in the third trimester of pregnancy, which is dysregulated earlier in GDM cases, is mostly supported by the increase in their plasmatic concentrations in GDM patients [51]. The input of specific tissues, like placental or periodontal tissues, to the pro-inflammatory state in GDM pregnancies is difficult to establish while isolating other sources. Our research group is working on in vitro and in vivo models for this purpose, targeting locally produced extracellular vesicles (EVs) as the main carriers of pro-inflammatory mediators (unpublished data). The signals and vehicles that regulate the secretion of pro-inflammatory mediators in GDM are not elucidated, which may help to understand the development of pro-inflammatory processes in GDM and its association to other inflammatory diseases like periodontitis.

## 6. Extracellular Vesicles in the Cross-Talk between Periodontal Tissues and the Placenta 

The International Society for Extracellular Vesicles suggested defining EVs as a generic term that refers to nanometer-scale particles naturally released from the cell that are delimited by a lipid bilayer and cannot replicate [92]. EVs can be classified into micro-vesicles, exosomes, and apoptotic bodies [93,94]. EVs secreted by one cell type are origin-specific and can be captured by other cell types and transfer their contents [92,93]. Among their multiple functions, EVs are part of the intercellular communication process, and they activate signaling pathways and contain different types of molecules, including receptors, lipids, proteins, nucleic acids (long non-coding RNAs, miRNAs, etc.), cytokines, fragments of organelles, and other cellular components [95,96]. EVs are involved in the pathogenesis of several chronic inflammatory and infectious diseases [94,97]. Furthermore, placental-derived EVs (PdEVs) in the circulating bloodstream are augmented in approximately 40% of pathological pregnancies, such as GDM and PE, compared with healthy pregnancies [98].

EVs represent a potential novel regulating mechanism involved in the modulation of maternal glucose homeostasis during pregnancy [99,100,101]. The increase in EV secretion or their cargo modification in a hyperglycemic status indicates that the diabetic environment could alter EVs’ bioactivity in target cells [102]. Moreover, placental-derived EVs could increase the release of TNF-α from endothelial cells [102,103,104]. Interestingly, circulating levels of TNF-α have been shown to be positively correlated to maternal insulin resistance, suggesting that placental EVs may modulate insulin sensitivity during gestation [102,103,104]. Additionally, the overexpression of TNF-α in GDM placentas was associated with a significant modification in the microRNA (miRNA) content of placental EVs [105].

An emerging potential pathway via EVs as communicators between the periodontium and the placenta, in the context of increased systemic inflammation, was proposed by our research group [106]. Despite the fact that there is a paucity of research about EVs obtained from periodontal tissues, our research group has led the research in this field [107]. Our recent published paper showed that the total concentration of EVs in GCF was increased in periodontitis compared with the healthy and gingivitis patients (*p* = 0.017) [108]. We observed that the total concentration of micro-vesicles and CD63 exosome markers was higher in patients with periodontitis compared with the healthy and gingivitis groups (*p* < 0.001 for both) [108]. The total concentration of EVs was correlated with the periodontal clinical parameters, like BOP (rho = 0.63, *p* = 0.002), PPD (rho = 0.56, *p* = 0.009), and CAL (rho = 0.48, *p* = 0.030) [108]. Moreover, we observed that the concentration of EVs isolated from the GCF of pregnant women during early pregnancy was significantly higher in the group of pregnant women who later developed GDM (*p*
**=** 0.0026), and the AUC-ROC analysis model of EVs as predictive markers of GDM had an almost perfect test result of 0.81 [107]. These results demonstrate that a concentration of 1.17 × 10^10^ particles/mL in GCF can be used as a cut-off point for GDM prediction with 63.3% sensitivity and 95.7% specificity [107]. Whereas our studies were focused on the identification of early biomarkers of GDM, they emphasized the role of GCF-EVs as speakers between periodontal inflammation and GDM. In the same sense, the concentration of EVs in GCF is increased in periodontitis and GDM pregnancies, and they are known to have a role in systemic inflammation activation in several pathologies. Thus, the content and bioactivity of EVs may potentially explain the molecular mechanisms that link periodontitis and GDM (Figure 2).

### EV Release and Bioactivity in GDM

During pregnancy, the delivery of placental-derived extracellular vesicles (PdEVs) into maternal circulation can be detected, since the 6th week of pregnancy and their concentration increases along with the progression of the pregnancy [104]. Existing evidence suggests that PdEVs may have a role in the maternal inflammation and vascular dysfunction observed in GDM pathophysiology [98]. In vitro studies have shown that trophoblast cells respond to a high glucose stimulus, increasing their EV release concentration and changing their bioactivity over endothelial cells, in which they raise the secretion of pro-inflammatory cytokines and potentially contribute to endothelial dysfunction [102]. Likewise, EVs from human umbilical vein endothelial cells (HUVECs) cultured under hyperglycemic conditions generated endothelial dysfunction in euglycemic HUVEC cells [99]. Moreover, EVs obtained from GDM pregnancies increased by 3.3-fold the release of pro-inflammatory cytokines from endothelial cells upon co-culture [24]. These results were ratified by Nakahara et al., who investigated the plasmatic concentrations of PdEVs from GDM, PE, and healthy pregnant woman and observed a 2.1-fold higher concentration of PdEVs in these pathological conditions [109]. Consistently, James-Allan et al. observed that mice treated with EVs from GDM pregnant women developed glucose intolerance (*p* = 0.006) [101]. However, the paracellular effects of totally circulating EVs in GDM are probably not unidirectional and may be explained by the release of EVs from other tissues, like the adipose or periodontal tissues, into placental tissues where they may promote insulin resistance in pregnant women.

As mentioned before, EVs usually contain miRNAs, and their structure helps to protect miRNAs from exogenous ribonucleases, allowing them to reach long-distance tissues [110]. miRNAs are non-coding RNAs that alter the phenotype of the target cells because they positively or negatively control gene expression and cellular mechanisms by binding to the 3’-untranslated region (UTR) of mRNAs, altering their stability or translation [111]. In the case of GDM, plasmatic EVs may contain specific miRNAs [105,111,112]. A possible cellular mechanism that can explain the variation in EV content and bioactivity in the context of hyperglycemia is the modifications in the heterogeneous nuclear riboprotein AA2B1 (hnRNPA2B1), which controls the loading of miRNAs into EVs. Specifically, the sumoylation of AA2B1 in exosomes may trigger a higher intracellular calcium concentration in response to hyperglycemia, altering its function [102]. Another explanation for the differential miRNA content is the differential expression in the GDM context of proteins involved in the biogenesis of miRNAs, such as Dicer or Drosha. For instance, there was a higher expression of mRNA of Drosha and Dicer in the peripheral blood from GDM patients compared with healthy pregnant controls (*p* < 0.001) [112].

## 7. Conclusions

Epidemiological studies support an association between periodontitis and GDM. However, the biological mechanisms and the related molecules involved in this association are primarily unknown. Until now, most research has focused on the effect of periodontal inflammation on the increased risk of pregnancy complications, such as preterm birth or PE. Little is known about the impact of periodontitis on placental inflammatory activation and its role in GDM development. Likewise, we do not have evidence about the role of GDM in exacerbating periodontal inflammation during pregnancy. Do both diseases share the systemic inflammation signature in their pathogenesis? As discussed above, sustained hyperglycemia during GDM pregnancies promotes the release of PdEVs, which contributes to a pro-inflammatory status. Similarly, periodontitis also contributes to the systemic inflammatory status, which is also related to the pathophysiology of GDM and makes this association biologically plausible.

The relationship between periodontitis and GDM may be mediated via direct (bacteria, their virulence factors, and EV translocation) or indirect mechanisms (pro-inflammatory mediators and EVs) (Figure 1). Our proposed model includes bidirectional cross-talk between the inflamed periodontal tissues and the placenta during pregnancy, with an increase of systemic inflammation and positive feedback between both conditions. In GDM conditions, hyperglycemia, oxidative stress, and the increase in the release of placental-derived extracellular vesicles (PdEVs) and placental hormones, like placental growth factor (PlGF) and placental lactogen (hPL), contribute to periodontal inflammation. In the other part, periodontitis, through bacteria or their virulence factor translocation, act like a constant challenge to immune inflammatory cells, increasing the release of pro-inflammatory mediators (CRP, TNF-α, IL-6, and IL-1β) and periodontal-derived EVs into circulation. These mechanisms could contribute to GDM development and enhance a pro-inflammatory environment in the placental tissue. A “vicious cycle” could be explained by the cross-talk between periodontitis and GDM, through distant local and systemic inflammation activation (Figure 3).

## Figures and Tables

**Figure 1 ijms-22-11831-f001:**
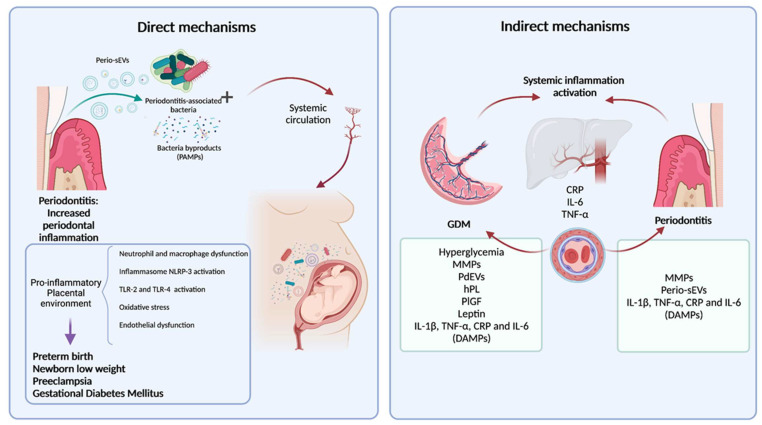
Direct and indirect mechanisms involved in the link between Gestational Diabetes Mellitus (GDM) and Periodontitis. A pro-inflammatory state contributes to GDM development and higher periodontitis severity. Among the direct mechanisms, periodontal bacteria or their byproducts are released into systemic circulation and can translocate into placental tissues, increasing the possibility of placental inflammation. Indirect mechanisms refer to the increase in pro-inflammatory mediators from periodontal pockets into circulation, reaching the liver and placenta, and vice-versa. Regarding extracellular vesicles (EVs), they may act via direct or indirect mechanisms, contributing to the cross-talk of both diseases. Perio-sEVs: periodontal-derived small extracellular vesicles; PAMPs: pathogen-associated molecular patterns; IL-1β: interleukin 1 beta; TNF-α: tumor necrosis factor alpha; CRP: C-reactive protein; IL-6: interleukin 6; DAMPs: damage-associated molecular patterns; TLR-2: toll-like receptor 2; TLR-4: toll-like receptor 4; NLRP-3: NOD-, LRR-, and pyrin domain-containing protein 3; MMPs: metalloproteinases; PdEVs: placental derived extracellular vesicles; hPL: human lactogen placental; PlGF: placental growth factor; Perio-EVs: periodontal-derived extracellular vesicles.

**Figure 2 ijms-22-11831-f002:**
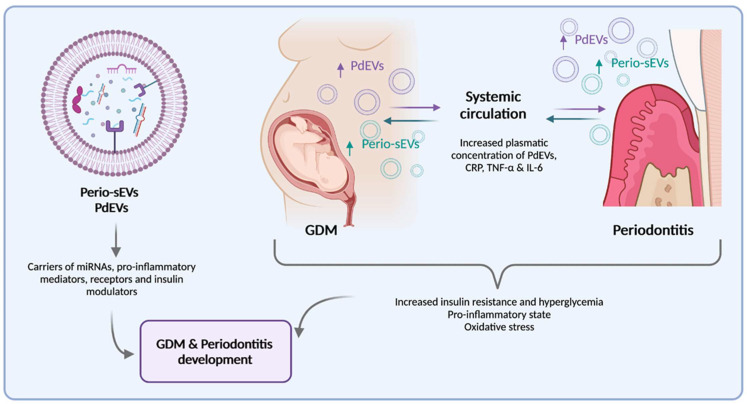
Role of periodontal-derived extracellular vesicles (Perio-sEVs) and placenta-derived extracellular vesicles (PdEVs) in Gestational Diabetes Mellitus (GDM) development. PdEVs are increased in plasma from GDM pregnant women along with C-reactive protein (CRP), tumor necrosis factor-alpha (TNF-α), and interleukin 6 (IL-6). PdEVs act as signal carriers between different cell types, and their cargo and bioactivity are also modified in GDM conditions. Overexpression or underexpression of certain microRNAs (miRNAs) can generate epigenetic modifications in recipient cells. A pro-inflammatory state can also be maintained by the release of PdEVs into the circulation and gingival crevicular fluid (GCF), which may aggravate periodontitis severity and contribute to the onset of GDM.

**Figure 3 ijms-22-11831-f003:**
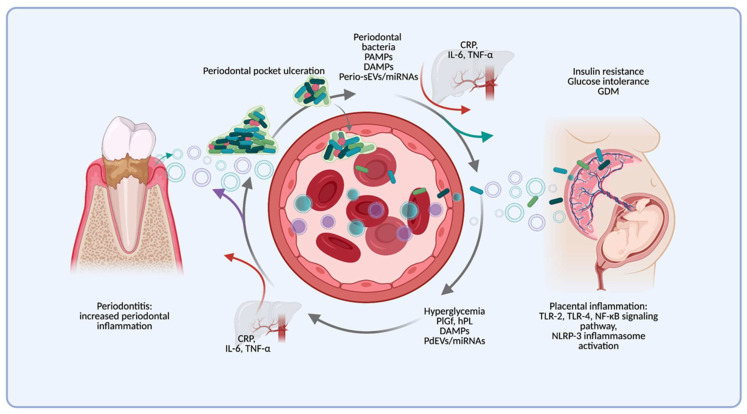
Proposed model of the association between Periodontitis and Gestational Diabetes Mellitus (GDM). In GDM conditions, hyperglycemia, oxidative stress, and placental-derived extracellular vesicles (PdEVs) contribute to a systemic and periodontal pro-inflammatory state and may directly affect periodontitis progression. Periodontitis, through bacterial translocation, pro-inflammatory mediators, and periodontal-derived sEVs (Perio-sEVs) released into systemic circulation, may contribute to GDM development over systemic inflammation and generate direct effects in the placenta. The low-grade systemic inflammation activation is mediated by the acute phase response in the liver and pro-inflammatory cytokines, which can be triggered by both diseases and alter their development. Thus, GDM and periodontitis are biologically connected in a “vicious cycle” that generates a cross-link via systemic inflammation activation. PAMPs: pathogen-associated molecular patterns; DAMPs: damage-associated molecular patterns; Perio-sEVs: periodontal-derived small extracellular vesicles; miRNA: microRNA; CRP: C-reactive protein; IL-6: interleukin 6; TNF-α: tumor necrosis factor alpha; TLR-2: toll-like receptor 2; TLR-4: toll-like receptor 4; NF-kB: nuclear factor kappa B; NLRP3: NOD-, LRR-, and pyrin domain-containing protein 3; IL-1β: interleukin 1 beta; MMPs: metalloproteinases; PdEVs: placental-derived extracellular vesicles; hPL: human lactogen placental; PlGF: placental growth factor; Perio-EVs: periodontal-derived extracellular vesicles.

**Table 1 ijms-22-11831-t001:** Association between Gestational Diabetes Mellitus onset, Periodontitis and Preeclampsia.

Disease	Parameters	Targets	Experimental Design	Statistical Index	Value (Cases vs. Controls)	(95% CI)	*p*-Value	References
GDM	(a) GI (b) PI (c) TMD (d) PPD (e) BOP	GDM (n = 65) and euglycemic pregnant women (n = 331)	Case–control	Mean values	(a) 1.98 vs. 0.86 (b) 2.65 vs. 0.62 (c) 1.15 vs. 0.84 (d) 3.92 vs. 2.20 (e) 56% and 7% *GDM vs. controls		All parameters *p* < 0.001	[62]
GDM	(a) PPD (b) CAL (c) BOP (d) Periodontitis and GDM association	GDM (n = 50) and euglycemic pregnant women (n = 50)	Case–control	Mean values and aOR	(a) 2.4 v/s 2.1 (b) 1.4 v/s 0.9 (c) 83.2% v/s 60.2 %. (d) aOR: 7.92 *GDM vs. controls	aOR: 1.66–37.70	(a)*p* = 0.02 (b)*p* = 0.003 (c) *p* = 0.001 (d) not reported	[63]
GDM and PE	(a) GDM incidence (b) Periodontitis and GDM association (c) Periodontitis and PE association (d) Periodontitis, GDM, and PE association	Periodontitis (n = 148), gingivitis (n = 184) and periodontally healthy pregnant women (n = 252)	Case–control	Mean values and aHR	(a) 19.6% v/s 4.4% (b) aHR: 4.12 (c) aHR: 2.20 (d) aHR: 18.79 *Periodontitis vs. controls	Periodontitis and GDM: 2.05–8.29 Periodontitis and PE: 0.86–5.6 Periodontitis, GDM, and PE: 7.45–47.4	(a,b) *p* = 0.001 (c) *p* = 0.01 (d) *p =* 0.001	[64]
GDM	(a) Periodontitis and GDM association in overall studies (b) Periodontitis and GDM association in high-quality studies	Cases (n = 624) and controls (n = 5724) in all studies Cases (n = 380) and controls (n = 1176) in high-quality studies	SR with MA	OR aOR	(a) OR: 1.66 (b.1) OR: 1.85 (b.2) aOR: 2.08	(a) 1.17–2.36 (b.1) 1.03–3.32 (b.2) 1.21–3.58	(a,b.1) *p* < 0.05 (b.2) *p* = 0.009	[65]
GDM	(a) Periodontitis and GDM association in cross-sectional studies (b) Periodontitis and GDM association in case–control studies	(a) GDM (n = 314) and controls (n = 4344) (b) GDM (n = 193) and controls (n = 396)	SR with MA	OR	(a) 1.67 (b) 1.69	(a) 1.2–2.32 (b) 0.68–4.21	Not reported	[67]
GDM	(a) BOP (b) PPD (c) CAL (d) PISA	GDM (n = 14) and cohort healthy pregnancies (n = 198)	Nested Case–control	Mean values	(a) 85.5 vs. 60% (b) 3.05 vs. 2.7 (c) 2.5 vs. 2.05 (d) 1548.9 vs. 877 *GDM vs. healthy cohort		(a,c,d) *p* < 0.001 (b) *p =* 0.002	[68]
Periodontitis	(a) Periodontitis prevalence (b) Periodontitis and GDM association (c) Periodontitis and past GDM association	Pregnant women (n = 256) with or without GDM, and non-pregnant women (n = 4234)	Case–control	Mean prevalence and aOR	(a) 44.8% vs. 13.2% (b) OR: 2.54 and aOR: 2.00 (c) OR: 2.07 *GDM vs. healthy pregnant woman	(b) OR: 0.94–6,86 and aOR: 0.65–6.2 (c) GDM OR: 0.67–6.37	Not reported	[69]
GDM	(a) Periodontitis, current DM, and past GDM association	Women who had histories of GDM (n = 113) and women without GDM histories (n = 4131) and current DM	Case–control	OR	(a) 8.7	(a) 2.5–29.8	Not reported	[70]

GI: gingival index; PI: plaque index; TMD: tooth mobility degree; PPD: periodontal pocket depth; BOP: bleeding on probing; CAL: clinical attachment loss; OR: odds ratio; aOR: adjusted odds ratio; aHR: adjusted hazards ratio; SR: systematic review; MA: meta-analysis; PISA: periodontal inflamed surface area; DM: diabetes mellitus.

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
