# Peer review of "Periodontitis and Gestational Diabetes Mellitus: A Potential Inflammatory Vicious Cycle"

_ijms, 2021, doi:10.3390/ijms222111831_

Round 1

Reviewer 1 Report

A well researched, well illustrated and largely written paper. Only minor criticisms. Many of the sentences are very long and would be more readable if broken up into sentences that include a single idea. Some seem to lack the correct wording or are incomplete such as "Moreover, the risk for PE increased with both, periodontitis and GDM in 20-fold more (HR = 19.96, 95% CI: 7.62-52.31, p-value = 0.001) [66–68]". Maybe a grammar check of the manuscript would provide a simple solution. You quote the "Dysbiotic theory" but I feel that the "Dysbiotic hypothesis" would be more appropriate.

Reviewer 2 Report

[General]

The authors reviewed the association between periodontal disease and gestational diabetes mellitus (GDM) from the wide points such as the epidemiology, proinflammatory molecules , and extracellular vesicles (EVs). Particularly, the role of EVs from placenta and periodontal tissue in GDM onset is of interest. Therefore, this review is meaningful and beneficial to the readers. 

However, several improvements are needed as described below.

[Major points]

  1. The results of epidemiological studies are important to judge the association between various factors and disease onset. The tabulation of these data is easy to understand for readers. Therefore, please tabulate the results of epidemiological studies. Candidate titles are as follows. Table 1 Association between GDM onset and periodontal disease and preeclampsia (line 184-202). Table 2 Association between obesity and periodontal disease  (Line 184-202).

[Minor points]

  1. (Line 351-352) I think that positive/negative predictive values (PPV/NPV) are unnecessary in review because PPV/NPV are affected by prevalence. 
  1. Please change “p-value = ***” to “p = ***” because the latter term is more general. Moreover, too small P value is not necessary. “P < 0.001” is sufficient.
  1. typo: unnecessary hyphen should be removed (periodonti-tis in line 18, 21, and 29; sus-ceptible in line 19; in-tercellular in line 31; pre-eclampsia in line 74; preg-nancies in line 101).
  1. (References) Numbers before authors should be deleted.
